# OpenReview forum: "MGDA-Decoupled: Geometry-Aware Multi-Objective Optimisation for DPO-based LLM Alignment"
_ICLR.cc/2026/Workshop/AFAA — AFAA 2026 Poster_

### Official Review · Reviewer_XkcY · 2026-02-20
**Review of "  MGDA-Decoupled: Geometry-Aware Multi-Objective Optimisation for DPO-based LLM Alignment"**

**Rating:** 3
**Confidence:** 3

**Summary:**

This paper proposes MGDA-DECOUPLED, a geometry-aware multi-objective optimization algorithm for LLM alignment. The core idea is to align LLMs across multiple conflicting objectives without requiring reinforcement learning or explicit reward models. The method finds a shared descent direction by computing mixing coefficients from loss-normalized gradients while applying them to raw gradients, explicitly accounting for the varying convergence dynamics of different objectives.

**Strengths:**

- The paper is technically well-motivated and clearly grounded in multi-objective optimization theory. By normalizing gradients by their corresponding loss values, the method assigns higher weight to objectives estimated to be farther from convergence, providing a principled mechanism for preventing easier objectives from dominating training
- The method avoids the computational overhead of RL pipelines and auxiliary reward models, making it a practical feature for fine-tuning workflows
- Separating the computation of mixing coefficients from the gradient aggregation step is a clean and well-motivated contribution. The geometric interpretation is intuitive
- The toy example in Figure 3 is effective. It isolates the failure modes of each baseline and provides a clear explanation for MGDA-DECOUPLED's faster convergence

**Weaknesses:**

- The empirical evidence does not demonstrate consistent superiority across objectives. In Table 1 (Gemma-2-2b-it), GROUPDRO achieves the highest net win rate on Honesty, and margins between most methods are often below one percentage point. For Qwen2.5-0.5B-Instruct (Table 2) MGDA-NORMALISED matches or exceeds MGDA-DECOUPLED on three out of four individual objectives, with MGDA-DECOUPLED holding only a marginal aggregate advantage (+4.7% vs. +4.1%). The paper acknowledges this as methods gravitating toward distinct non-dominated stationary points but this framing also weakens the claim that MGDA-DECOUPLED provides systematically superior trade-offs rather than just a different point on the Pareto front
- The robustness of the reported gains is difficult to assess. No results are averaged across random seeds, and no confidence intervals are reported beyond the paired bootstrap test against the REFERENCE baseline. Given that the LLM Judge evaluation pipeline can introduce non-trivial noise, it is unclear whether the small observed differences between methods are stable across runs
- The experimental setting may structurally limit the method's advantages. The authors themselves note that UltraFeedback objectives such as Honesty and Truthfulness are correlated rather than conflicting on many prompts, and that the DPO KL-divergence term artificially aligns gradients across objectives by pulling all of them toward the reference model. Both factors dampen gradient conflict density which is the condition under which geometry-aware methods offer the most benefit. The case for MGDA-DECOUPLED would be substantially stronger in an explicitly adversarial trade-off setting, such as helpfulness versus safety under conflicting prompt distributions, where gradient opposition is structurally guaranteed

---

### Official Review · Reviewer_2eSf · 2026-02-21
**Geometry-Aware Multi-Objective DPO: Solid Alignment Method with Limited Evaluation Breadth**

**Rating:** 4
**Confidence:** 3

**Summary:**

The paper proposes MGDA-Decoupled, a geometry-aware multi-objective optimisation method for DPO-based LLM alignment. The approach dynamically computes gradient mixing coefficients to better balance competing alignment objectives (e.g., helpfulness, honesty, harmlessness), motivated by concerns that fixed scalarisation may introduce procedural unfairness. Experiments on the UltraFeedback dataset with two instruction-tuned models show consistent improvements in win rates over several baselines.

**Strengths:**

1. Well-motivated formulation of multi-objective alignment as a procedural fairness problem.
2. Technically sound and clearly described geometry-aware optimisation method.
3. Practical design that remains within the lightweight DPO framework.
4. Empirical results show consistent improvements across evaluated objectives.
5. Strong topical fit for the AFAA workshop.

**Weaknesses:**

1. Performance gains over baseline alignment methods are relatively modest.
2. Fairness analysis focuses on objective balancing rather than downstream bias or societal fairness.
3. Evaluation relies on LLM-as-a-judge, which introduces additional uncertainty.
4. Experiments are limited to a single dataset (UltraFeedback).
5. Only two base models are evaluated, leaving questions about generalizability across architectures and scales.

---

### Official Review · Reviewer_buMC · 2026-02-24
**Review of "MGDA-Decoupled: Geometry-Aware Multi-Objective Optimisation for DPO-based LLM Alignment"**

**Rating:** 4
**Confidence:** 2

**Summary:**

The paper introduces MGDA-Decoupled, a dynamic scalarization algorithm for multi-objective LLM alignment within Direct Preference Optimization framework. To prevent procedural unfairness and neglect of harder-to-optimize objectives, the authors adapt the Multiple Gradient Descent Algorithm (MGDA). The method is evaluated on Gemma-202b-it and Qwen2.5-0.5B-Instruct using the UltraFeedback dataset, showing significant improvements in net win rates against baselines.

**Strengths:**

* The 2D toy example (Figure 3) is helps to intuitively demonstrate the failure modes of standard MGDA, Uniform and GroupDRO when objectives have vastly different gradient magnitudes but share a global optimum.

* By operating entirely within the DPO paradigm, the method avoids the heavy computational overhead and instability associated with RLHF/PPO and explicit reward models.

* The authors tested their methods across two different models to check the effectiveness.

**Weaknesses:**

* The authors state that the models are trained for only 1 epoch. It is unclear if 1 epoch is sufficient for the training to converge, this raises a critical question of whether the observed performance are stable outcomes or some artifacts of early stopping.

* The GroupDRO baseline relies on a hyper parameter  η, for the toy example the authors set  η=0.01 and for training they set it to  η=0.1. There is no documented ablation or grid search to show that this is the optimal setting.

* To ensure valid minibatches, smaller datasets are oversampled. This means that smaller datasets are effectively trained for more than one epoch, creating an imbalance.

Minor Weakness:
* Formatting issues in the appendix, the formatting in Sample model responses are broken in several places. For instances, bullet points and bold texts are rendered incorrectly, example * ** It’s a sensitive subject:**.


The paper offers a well-reasoned, mathematically sound and effectively demonstrated improvement for multi-objective DPO. The theoretical justification is solid. I would recommend accepting this paper for the workshop.

---

### Meta-Review · Area_Chair_mcTp · 2026-02-26

**Recommendation:** Main Papers Track
**Confidence:** 5

**Metareview:**

The paper proposes MGDA-Decoupled, a geometry-aware multi-objective optimization method for DPO-based LLM alignment, and explicitly frames multi-objective alignment as a procedural fairness problem. Reviewers find the motivation compelling and the method technically sound and practical, with the toy example clearly illustrating baseline failure modes. Experiments on UltraFeedback with two models show improved aggregate win rates over several baselines. Key concerns focus on empirical robustness and scope: gains are sometimes modest and not uniformly strongest per-objective, evaluation relies on LLM-as-a-judge, and results lack multi-seed averaging/uncertainty estimates. Overall, with two accepts and one borderline, I lean toward acceptance.

---

### Decision · Program_Chairs · 2026-03-02

Accept (Poster)